# SUM-OF-PARTS MODELS: FAITHFUL ATTRIBUTIONS FOR GROUPS OF FEATURES

## ABSTRACT

An explanation of a machine learning model is considered *"faithful"* if it accurately reflects the model's decision-making process. However, explanations such as feature attributions for deep learning are not guaranteed to be faithful, and can produce potentially misleading interpretations. In this work, we develop *Sum-of-Parts (SOP)*, a class of models whose predictions come with grouped feature attributions that are faithful-by-construction. Specifically, we modify black-box models to have faithful interpretations, by aggregating predictions over varying subsets of features. We evaluate SOP on benchmarks with standard interpretability metrics, and in a case study, we use the faithful explanations from SOP to help astrophysicists discover new knowledge about galaxy formation.

## 1    INTRODUCTION

In many high-stakes domains like medicine, law, and automation, important decisions must be backed by well-informed and well-reasoned arguments. However, many machine learning (ML) models are not able to give explanations for their behaviors. One type of explanations for ML models is *feature attribution*: the identification of input features that were relevant to the prediction (Molnar, 2022).

For example, in medicine, ML models can assist physicians in diagnosing a variety of lung, heart, and other chest conditions from X-ray images (Rajpurkar et al., 2017; Chambon et al., 2022; Zhang et al., 2022; Termritthikun et al., 2023). However, physicians only trust the decision of the model if an explanation identifies regions of the X-ray that make sense (Reyes et al., 2020). Such explanations are increasingly requested as new biases are discovered in these models (Glocker et al., 2022).

The field has proposed a variety of feature attribution methods to explain ML models. One category consist of post-hoc attributions (Ribeiro et al., 2016; Lundberg & Lee, 2017; Petsiuk et al., 2018; Selvaraju et al., 2016; Sundararajan et al., 2017a), which have the benefit of being able to apply to any model. Another category of approaches instead build feature attributions directly into the model (Wiegreffe & Pinter, 2019; Jain et al., 2020; Simonyan et al., 2014a; Sabour et al., 2017; Courbariaux et al., 2015), which promise more accurate attributions but require specially designed architectures or training procedures.

However, feature attributions do not always accurately represent the model's prediction process, a property known as *faithfulness*. An explanation is said to be faithful if it correctly represents the reasoning process of a model (Lyu et al., 2022). For a feature attribution method, this means that the highlighted features should actually influence the model's prediction. For instance, suppose a ML model for X-rays uses the presence of a bone fragment to predict a fracture while ignoring a jagged line. A faithful feature attribution should assign a positive score to the bone fragment while assigning a score of zero to the jagged line. On the other hand, an unfaithful feature attribution would assign a positive score irrelevant regions. Unfortunately, studies have found that many post-hoc feature attributions do not satisfy basic sanity checks for faithfulness (Lyu et al., 2022).

In this paper, we first identify a fundamental barrier for feature attributions arising from the curse of dimensionality. Specifically, we prove that feature attributions incur exponentially large error in faithfulness tests for simple settings. These theoretical examples motivate a different type of attribution that scores *groups* of features to overcome this inherent obstacle. Motivated by these challenges, we develop Sum-of-Parts models (SOP), a class of models that attributes predictions to groups of features, which are illustrated in Figure 1. Our approach has three main advantages: SOP

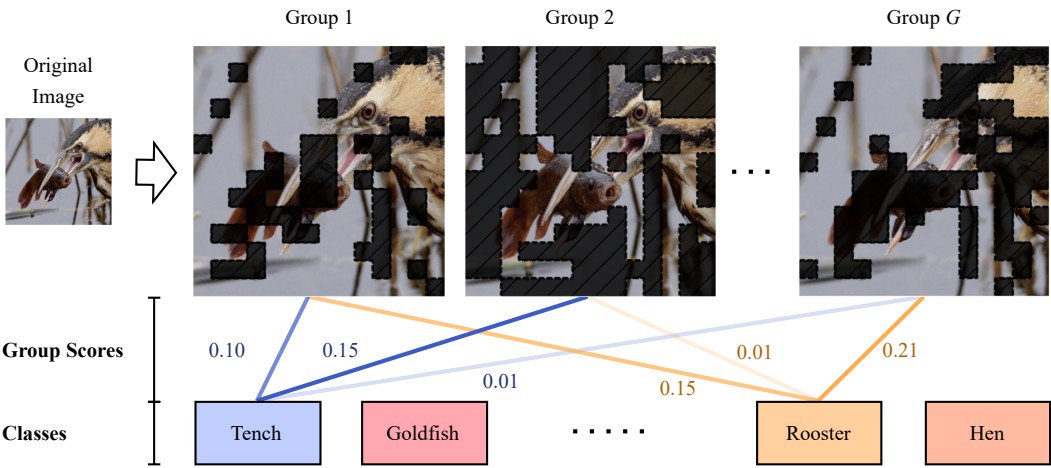

Figure 1: Visualization of grouped attributions. For a set of group attributions, scores are assigned to groups of features instead of individual features. Each group has a binary assignment in whether or not to include each feature. The score for each group represents how much each group of features together contributes to the prediction. We can see that masks can be interpreted as objects kept and objects removed. In this example, group 2, which includes the fish and the predator, contributes 15% to predicting "tench", while group $G$, which has the fish and dark lines removed, contributes only 1% to predicting "tench", but 21% to predicting "Rooster".

models (1) provide grouped attributions that overcome theoretical limitations of feature attributions; (2) are faithful by construction, avoiding pitfalls of post-hoc approaches; and (3) are compatible with any backbone architecture. Our contributions are as follows:

1. We prove that feature attributions must incur at least exponentially large error in tests of faithfulness for simple settings. We further show that grouped attributions can overcome this limitation.

2. We develop Sum-of-Parts (SOP), a class of models with group-sparse feature attributions that are faithful by construction and are compatible with any backbone architecture.

3. We evaluate our approach in a standard image benchmark ImageNet with interpretability metrics.

4. In a case study, we use faithful attributions of SOP from weak lensing maps and uncover novel insights about galaxy formation meaningful to cosmologists.

## 2 INHERENT BARRIERS FOR FEATURE ATTRIBUTIONS

Feature attributions are one of the most common forms of explanation for ML models. However, numerous studies have found that feature attributions fail basic sanity checks (Adebayo et al., 2018; Sundararajan et al., 2017b) and interpretability tests (Kindermans et al., 2019; Bilodeau et al., 2022).

Perturbation tests are a widely-used technique for evaluating the faithfulness of an explanation (Petsiuk et al., 2018; Vasu & Long, 2020b; DeYoung et al., 2020). These tests insert or delete various subsets of features from the input and check if the change in model prediction is in line with the scores from the feature attribution. We first formalize the error of a deletion-style test for a feature attribution on a subset of features.

**Definition 1.** *(Deletion error) The* deletion error *of an feature attribution* $\alpha \in \mathbb{R}^d$ *for a model* $f : \mathbb{R}^d \to \mathbb{R}$ *when removing a subset of features* $S$ *from an input* $x$ *is*

$$\text{DelErr}(\alpha, S) = \left| f(x) - f(x_{\neg S}) - \sum_{i \in S} \alpha_i \right| \quad \text{where} \quad (x_{\neg S})_j = \begin{cases} x_j & \text{if } j \notin S \\ 0 & \text{otherwise} \end{cases} \tag{1}$$

*The total deletion error is* $\sum_{S \in \mathcal{P}} \text{DelErr}(\alpha, S)$ *where* $\mathcal{P}$ *is the powerset of* $\{1, \ldots, d\}$.

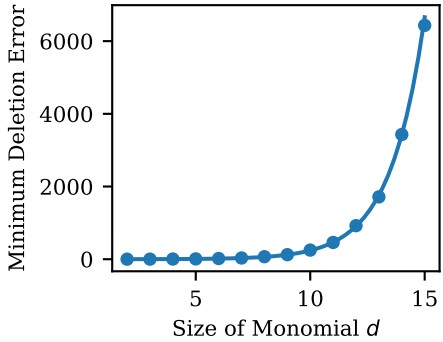 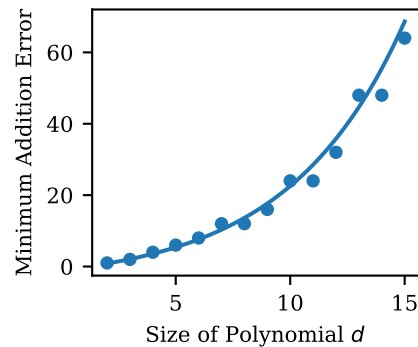

(a) Minimum deletion error for monomials. Fitted function: $\text{DelErr}(d) = e^{\gamma_1 d + \gamma_0}$ where $(\gamma_1, \gamma_0) = (0.664, -1.159)$.

(b) Minimum insertion error for binomials. Fitted function: $\text{InsErr}(d) = e^{\lambda_2 d + \lambda_1} + \lambda_0$ where $(\lambda_2, \lambda_1, \lambda_0) = (0.198, 1.332, 4.778)$.

Figure 2: The minimum (a) deletion error of monimoals of size $d$ and (b) insertion errors of binomials of size $d$, where the minimum is over all possible feature attributions. These lower bounds suggest an inherent fundamental limitation of feature attributions in faithfully explaining correlated features.

The deletion error measures how well the total attribution from features in $S$ aligns with the change in model prediction when removing the same features from $x$. Intuitively, a faithful attribution score of the $i$th feature should reflect the change in model prediction after the $i$th feature is removed and thus have low deletion error. We can formalize an analogous error for insertion-style tests as follows:

**Definition 2.** *(Insertion error) The* insertion error *of an feature attribution* $\alpha \in \mathbb{R}^d$ *for a model* $f : \mathbb{R}^d \to \mathbb{R}$ *when inserting a subset of features* $S$ *from an input* $x$ *is*

$$\text{InsErr}(\alpha, S) = \left| f(x_S) - f(0_d) - \sum_{i \in S} \alpha_i \right| \quad \text{where} \quad (x_S)_j = \begin{cases} x_j & \text{if } j \in S \\ 0 & \text{otherwise} \end{cases} \tag{2}$$

*The total insertion error is* $\sum_{S \in \mathcal{P}} \text{InsErr}(\alpha, S)$ *where* $\mathcal{P}$ *is the powerset of* $\{1, \dots, d\}$.

The insertion error measures how well the total attribution from features in $S$ aligns with the change in model prediction when adding the same features to the $0_d$ vector. Note that if an explanation is faithful, then it achieves low deletion and insertion error. For example, a linear model $f(x) = \theta^T x$ is often described as an interpretable model because it admits a feature attribution $\alpha_i = \theta_i x_i$ that achieves zero deletion and insertion error. Common sanity checks for feature attributions often take the form of insertion and deletion on specific subsets of features (Petsiuk et al., 2018).

## 2.1 FEATURE ATTRIBUTIONS INCUR A MINIMUM OF EXPONENTIAL ERROR

In this section, we provide two simple polynomial settings where any choice of feature attribution is guaranteed to incur at least exponential deletion and insertion error across all possible subsets. The key property in these examples is the presence of highly correlated features, which pose an insurmountable challenge for feature attributions. We defer all proofs to Appendix B, and begin with the first setting: multilinear monomials, or the product of $d$ Boolean inputs.

**Theorem 1** (Deletion Error for Monomials). *Let* $p : \{0, 1\}^d \to \{0, 1\}$ *be a multilinear monomial function of* $d \leq 20$ *variables,* $p(x) = \prod_{i=1}^d x_i$. *Then, there exists an* $x$ *such that any feature attribution for* $p$ *at* $x$ *will incur an approximate lower bound of* $e^{\gamma_1 d + \gamma_0}$ *total deletion error, where* $(\gamma_1, \gamma_0) = (0.664, -1.159)$.

In other words, Theorem 1 states that the total deletion error of any feature attribution of a monomial will grow exponentially with respect to the dimension, as visualized in Figure 2a. For high-dimensional problems, this suggests that there does not exist a feature attribution that satisfies all possible deletion tests. On the other hand, monomials can easily achieve low insertion error, as formalized in Lemma 1.

**Lemma 1** (Insertion Error for Monomials). *Let $p : \{0,1\}^d \rightarrow \{0,1\}$ be a multilinear monomial function of $d$ variables, $p(x) = \prod_{i=1}^{d} x_i$. Then, for all $x$, there exists a feature attribution for $p$ at $x$ that incurs at most $1$ total insertion error.*

However, once we slightly increase the function complexity to binomials, we find that the total insertion error of any feature attribution will grow exponentially with respect to $d$, as shown in Figure 2b. The two terms in the binomial must have some overlapping features or else the problem reduces to a monomial.

**Theorem 2** (Insertion Error for Binomials). *Let $p : \{0,1\}^d \rightarrow \{0,1,2\}$ be a multilinear binomial polynomial function of $d$ variables. Furthermore suppose that the features can be partitioned into $(S_1, S_2, S_3)$ of equal sizes where $p(x) = \prod_{i \in S_1 \cup S_2} x_i + \prod_{j \in S_2 \cup S_3} x_j$. Then, there exists an $x$ such that any feature attribution for $p$ at $x$ will incur an approximate lower bound of $\exp(\lambda_2 d + \lambda_1) + \lambda_0$ error in insertion-based faithfulness tests, where $(\lambda_2, \lambda_1, \lambda_0) = (0.198, 1.332, 4.778)$ and $d \leq 20$.*

In combination, Theorems 1 and 2 imply that even for simple problems (Boolean monomials and binomials), the total deletion and insertion error grows exponentially with respect to the dimension.[1] This is precisely the curse of dimensionality, but for feature attributions. These results suggest that a fundamentally different attribution is necessary in order to satisfy deletion and insertion tests.

## 2.2 GROUPED ATTRIBUTIONS OVERCOME BARRIERS FOR FEATURE ATTRIBUTIONS

The inherent limitations of feature attributions stems from the highly correlated features. A standard feature attribution is limited to assigning one number to each feature. This design is fundamentally unable to accurately model interactions between multiple features, as seen in Theorems 1 and 2.

To explain these correlated effects, we explore a different type of attributions called *grouped attributions*. Grouped attributions assign scores to groups of features instead of individual features. In a grouped attribution, a group only contributes its score if all of its features are present. This concept is formalized in Definition 3.

**Definition 3.** *Let $x \in \mathbb{R}^d$ be an example, and let $S_1, \ldots, S_G \in \{0,1\}^d$ designate $G$ groups of features where $j \in S_i$ if feature $j$ is included in the $i$th group. Then, a grouped feature attribution is a collection $\beta = \{(S_i, c_i)\}_{i=1}^{G}$ where $c_i \in \mathbb{R}$ is the attributed score for the $i$th group of features $S_i$.*

Grouped attributions have three main characteristics. First, unlike standard feature attributions, a single feature can show up in multiple groups with different scores. Second, the standard feature attribution is a special case where $S_i$ is the singleton set $\{i\}$ for $i = 1, \ldots, G$ for $G = d$. Third, there exists grouped attributions that can succinctly describe the earlier settings from Theorems 1 and 2 with zero insertion and deletion error (Corollary 1 in Appendix B).

To summarize, grouped attributions are able to overcome exponentially growing insertion and deletion errors when the features interact with each other. In contrast, traditional feature attributions lack this property on even simple settings.

## 3 SUM-OF-PARTS MODELS

In this section, we develop the Sum-of-Parts (SOP) framework, a way to create faithful grouped attributions. Our proposed grouped attributions consist of two parts: the subsets of features called groups $(S_1, \ldots, S_G) \in [0,1]^d$ and the scores for each group $(c_1, \ldots, c_G)$. We divide our approach into two main modules: GroupGen which generates the groups $S_i$ of features from an input, and GroupSelect which assigns scores $c_i$ to select which groups to use for prediction, as in Figure 3.

These groups and their corresponding scores form the grouped attribution of SOP. To make a prediction our approach linearly aggregates the prediction of each group according to the score to produce a final prediction. Since the prediction for a group solely relies on the features within the group, the grouped attribution is faithful-by-construction to the prediction.

---

[1]The proof technique for Theorems 1 and 2 involves computing a verifiable certificate at each $d$. We were able to computationally verify the result up to $d \leq 20$, and hence the theorem statements are proven only for $d \leq 20$. We conjecture that a general result holds for $d > 20$ for both the insertion and deletion settings.

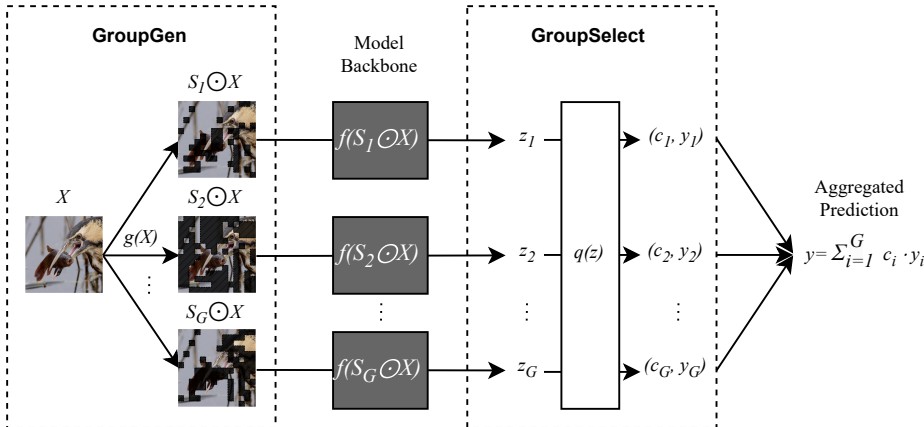

Figure 3: Structure of a Sum-of-Parts Model. A group generator $g$ first generates groups of features. Each group of features $S_i \odot X$ is then passed through the black-box model to obtain the group embedding $z_i$. A group selector $q$ then assigns a score $c_i$ to each group $i$'s representation. The partial logits are then aggregated with a weighted sum to get the predicted logit $y$ for a class.

**Group Generator.** The group generator $\mathsf{GroupGen} : \mathbb{R}^d \to [0,1]^{G \times d}$ takes in an input $X \in \mathbb{R}^d$ and outputs $G$ masks, each of which corresponds to a group $S_i \in [0,1]^d$. To generate these masks, we use a self-attention mechanism (Vaswani et al., 2017) to parameterize a probability distributions over features. The classic attention layer is

$$\mathsf{Attention}(X) = \mathsf{softmax}\left(\frac{W_q X (W_k X)^T}{\sqrt{d_k}}\right) W_v X$$

where $W_q, W_k, W_v$ are learned parameters.

However, the outputs of self attention are continuous and dense. Furthermore, we only need the attention weights to generate groups and can ignore the value. To make groups interpretable, we use a sparse variant using the sparsemax operator (Martins & Astudillo, 2016) without the value:

$$\mathsf{GroupGen}(X) = \mathsf{sparsemax}\left(\frac{W_q X (W_k X)^T}{\sqrt{d}}\right) \tag{3}$$

where $W_q, W_k \in \mathbb{R}^d$. The SparseMax operator uses a simplex projection to make the attention weights sparse. In total, the generator computes sparse attention weights and recombines the input features into groups $S_i$.

**Group Selector.** After we acquire these groups, we use the backbone model $f : \mathbb{R}^d \to \mathbb{R}^h$ to obtain each group's encoding $z_i = f(S_i \odot X)$ with embedding dimension $h$, where $\odot$ is Hadamard product. The goal of the second module, $\mathsf{GroupSelect}$, is to now choose a sparse subset of these groups to use for prediction. Sparsity ensures that a human interpreting the result is not overloaded with too many scores.

The group selector $\mathsf{GroupSelect}$ takes in the output of the backbone from all the groups $z_1, \ldots, z_G \in \mathbb{R}^h$ and produces scores $(c_1, \ldots, c_G) \in [0,1]^G$ and logits $(y_1, \ldots, y_G) \in \mathbb{R}^G$ for all groups. To assign a score to each group, we again use a modified sparse attention

$$\mathsf{GroupSelect}(z_1, \ldots, z_G) = \mathsf{sparsemax}\left(\frac{W_{q'} C (W_{k'} z)^T}{\sqrt{h}}\right), C z^T \tag{4}$$

where $W_{q'}, W_{k'}, C \in \mathbb{R}^h$. We use a projected class weight $W_{q'} C$ to query projected group encodings $W_{k'} z$. In practice, we can initialize the value weight $C$ to the weight matrix in the linear classifier of a pretrained model. $\mathsf{GroupSelect}$ then simultaneously produces the scores assigned to all groups $(c_1, \ldots, c_G)$ and each group's partial prediction $(y_1, \ldots, y_G)$.

The final prediction is then made by $y = \sum_{i=1}^{G} c_i y_i$, and the corresponding group attribution is $(c_1, S_1), \ldots,$

|  |  | LIME | SHAP | RISE | Grad-CAM | IntGrad | Archi-pelago | FRESH | SOP (ours) |
|---|---|---|---|---|---|---|---|---|---|
| ImageNet | Perf $\uparrow$ |  |  | —0.9160— |  |  |  | 0.8560 | 0.8880 |
|  | Ins $\uparrow$ | 0.5121 | 0.6130 | 0.5816 | 0.4545 | 0.3232 | 0.5499 | 0.5979 | **0.6149** |
|  | Ins$_G$ $\uparrow$ | 0.6121 | 0.6254 | 0.6180 | 0.6303 | 0.4909 | **0.6472** | 0.6195 | 0.6408 |
|  | Del $\downarrow$ | 0.3798 | 0.3009 | 0.4066 | 0.4532 | **0.2357** | 0.3620 | 0.4132 | 0.3929 |
|  | Del$_G$ $\downarrow$ | 0.3254 | 0.3008 | 0.3135 | 0.3104 | 0.5612 | 0.3064 | 0.3302 | **0.2973** |

Table 1: Results on ImageNet on all baselines and SOP on accuracy, insertion, grouped insertion, deletion, and grouped deletion. If a metric has $\uparrow$, it means higher numbers in the metric is better, and vice versa. For accuracy, post-hoc methods show the accuracy of the original model.

$(c_G, S_G)$. Since we use a sparsemax operator, in practice there can be significantly fewer than $G$ groups that are active in the final prediction. This group attribution is faithful to the model since the prediction uses exactly these groups $S_i$, each of which is weighted precisely by the scores $c_i$. As we are "summing" weighted "parts" of inputs, we call this a Sum-of-Parts model, the complete algorithm of which can be found in Algorithm 1.

## 4 EVALUATING SOP GROUPED ATTRIBUTIONS

In this section, we perform a standard evaluation with commonly-used metrics for measuring the quality of a feature attribution. These metrics align with the insertion and deletion error analyzed in Section 2. We find that our grouped attributions can improve upon the majority of metrics over standard feature attributions, which is consistent with our theoretical results.

### 4.1 EXPERIMENTAL SETUPS

We evaluate SOP on ImageNet (Russakovsky et al., 2015) for single-label classification. We use Vision Transformer (Dosovitskiy et al., 2021) as our backbone. More information about training and datasets are in Appendix D.1.

We compare against different types of baselines:

1. *Surrogate-model-based*: LIME (Ribeiro et al., 2016), SHAP (Lundberg & Lee, 2017)

2. *Perturbation-based*: RISE (Petsiuk et al., 2018)

3. *Gradient-based*: GradCAM (Selvaraju et al., 2016), IntGrad (Sundararajan et al., 2017a)

4. *Built-in explanation*: FRESH (Jain et al., 2020)

To evaluate our approach, we use interpretability metrics that are standard practice in the literature for feature attributions (Petsiuk et al., 2018; Vasu & Long, 2020a; Jain et al., 2020). We summarize these metrics as follows and provide precise descriptions in Appendix D.2:

1. **Accuracy:** We measure the standard accuracy of the model. For methods that build explanations into the model such as SOP, it is desirable to maintain good performance.

2. **Insertion and Deletion:** We measure faithfulness of attributions on predictions with insertion and deletion tests that are standard for feature attributions (Petsiuk et al., 2018). These tests insert and delete features pixel by pixel.

3. **Grouped Insertion and Deletion:** Insertion and deletion tests were originally made for standard feature attributions, which assign at most one score per feature. Grouped attributions can have multiple scores per feature if a feature shows up in multiple groups. We therefore generalize these tests to their natural group analogue, which inserts and deletes features in groups.

## 4.2 RESULTS AND DISCUSSIONS

**Accuracy.** To evaluate the performance of built-in explanation models have, we evaluate on accuracy. The intuition is that built-in attributions use a subset of features when they make the prediction. Therefore, it is possible that they do not have the same performance as the original models. A slight performance drop is an acceptable trade-off, while a large drop makes the model unusable.

We compare with FRESH which is also a model with built-in attributions that initially works for language but we adapt for vision. Table 1 shows that SOP retains the most accuracy on ImageNet and no less than FRESH. This shows that our built-in grouped attributions do not degrade model performance while adding faithful attributions. The multiple groups are potentially the advantage of SOP over single-group attributions from FRESH to model interactions between different groups of features.

**Insertion and Deletion.** To evaluate how faithful the attributions are, we evaluate on insertion and deletion tests. The intuition behind insertion is that, if the attribution scores are faithful, then adding the highest scored features first from the blank image will give a higher AUC, and deleting them first from the full image will give a low AUC. While Petsiuk et al. (2018) perturb an image by blurring to avoid adding spurious correlations to the classifier, this may not entirely remove a feature. Since modern backbones (such as the Vision Transformer that we use) are known to not be as biased as classic models when blacking out features (Jain et al., 2022), we simply replace features entirely with zeros which correspond to gray pixels. Also, to accommodate the tests designed for individual-feature attributions, we first perform weighted sum on the groups of features from SOP and then do insertion and deletion tests on the aggregated attributions.

We compare against all the post-hoc and built-in baselines. Table 1 shows that SOP has the best insertion AUC among all methods for ImageNet. Having higher insertion scores shows that the highest scored attributions from SOP are more sufficient than other methods in making the prediction. While the deletion scores are lower, SOP does not promise that the attributions it selects are comprehensive, and thus have the potential of lowering the deletion scores.

**Grouped Insertion and Deletion.** While we can still technically evaluate grouped attributions with pixel-wise insertion and deletion tests, it does not quite match the semantics of a grouped attribution, which score groups of features instead of individual features. A standard feature attribution method scores individual pixels, and therefore classic tests check whether inserting and deleting pixels one at a time aligns with the scores. In contrast, grouped attributions assign scores for groups of features, and thus a grouped insertion and deletion test assesses whether deleting groups of features at a time aligns with the scores.

Table 1 shows that SOP outperforms all other baselines in grouped deletion, and all but Archipelago in grouped insertion, while still being close to Archipelago. This shows that SOP finds grouped attribution that are better at determining which groups of features contribute more to the prediction. This is to be expected as it is faithful-by-construction.

## 5 CASE STUDY: COSMOLOGY

While outperforming other methods on standard metrics shows the advantage of our grouped attributions, the ultimate goal of interpretability methods is for domain experts to use these tools and be able to use the explanations in real settings. To validate the usability of our approach, we collaborated with domain experts and used SOP to discover new cosmological knowledge about the expansion of the universe and the growth of cosmic structure. We find that the groups generated with SOP contain semantically meaningful structures to cosmologists. The resulting scores of these groups led to findings linking certain cosmological structures to the initial state of the universe, some of which were surprising and previously not known.

Weak lensing maps in cosmology calculate the spatial distribution of matter density in the universe using precise measurements of the shapes of $\sim 100$ million galaxies (Gatti et al., 2021). The shape of each galaxy is distorted (sheared and magnified) due to the curvature of spacetime induced by mass inhomogenities as light travels towards us. Cosmologists have techniques that can infer the distribution of mass in the universe from these distortions, resulting in a weak lensing map Jeffrey et al. (2021).

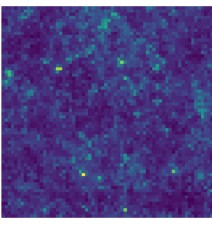 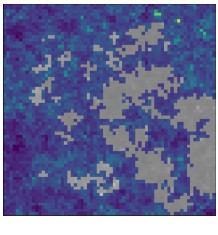 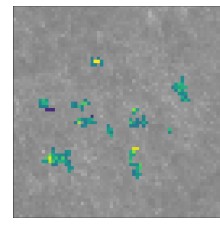

Weak Lensing Map               Void               Cluster

Figure 4: A weak lensing mass map (left) contains large dark areas which are voids, and hot pixels which are clusters. Voids (middle) are darker and larger areas in weak lensing maps. Clusters (right) are small groups of hot pixels. We find that voids are used more in predicting both $\Omega_m$ and $\sigma_8$ from such noiseless maps. Clusters are used less in general, but comparatively more for $\sigma_8$.

Cosmologists hope to use weak lensing maps to predict two key parameters related to the initial state of the universe: $\Omega_m$ and $\sigma_8$. $\Omega_m$ captures the average energy density of all matter in the universe (relative to the total energy density which includes radiation and dark energy), while $\sigma_8$ describes the fluctuation of matter distribution (see e.g. Abbott et al. (2022)). From these parameters, a cosmologist can simulate how cosmological structures, such as galaxies, superclusters and voids, develop throughout cosmic history. However, $\Omega_m$ and $\sigma_8$ are not directly measurable, and the inverse relation from cosmological structures in the weak lensing map to $\Omega_m$ and $\sigma_8$ is unknown.

One approach to inferring $\Omega_m$ and $\sigma_8$ from weak lensing maps, as demonstrated for example by Ribli et al. (2019); Matilla et al. (2020); Fluri et al. (2022), is to apply deep learning models that can compare measurements to simulated weak lensing maps. Even though these models have high performance, we do not fully understand how they predict $\Omega_m$ and $\sigma_8$. As a result, the following remains an open question in cosmology:

*What structures from weak lensing maps can we use to infer the cosmological parameters $\Omega_m$ and $\sigma_8$?*

In collaboration with expert cosmologists, we use convolutional networks trained to predict $\Omega_m$ and $\sigma_8$ as the backbone of an SOP model to get accurate predictions with faithful group attributions. Crucially, the guarantee of faithfulness in SOP provides confidence that the attributions reflect how the model makes its prediction, as opposed to possibly being a red herring. We then interpret and analyze these attributions and understand how structures in weak lensing maps of CosmoGridV1 (Kacprzak et al., 2023) influence $\Omega_m$ and $\sigma_8$.

**Cosmological findings:** Our initial findings come from grouped attributions that correspond to two known structures in the weak lensing maps (as identified by cosmologists): voids and clusters. Voids are large regions that are under-dense relative to the mean density and appear as dark regions in the weak lensing mass maps, whereas clusters are areas of concentrated high density and appear as bright dots. Figure 4 shows an example of voids (middle panel) and an example of clusters (right panel), both of which are automatically learned as groups in the SOP model without supervision. We use standard deviation $\sigma$ away from the mean mass intensity for each map to define voids and clusters, where voids are groups that have mean density $\leq 0$ and clusters are groups that have overdensity $\geq +3\sigma$. A precise definition of these structures is provided in Appendix E.

We summarize the discoveries that we made with cosmologists on how clusters and voids influence the prediction of $\Omega_m$ and $\sigma_8$ as follows:

1. A new finding of our work relates to the distinction between the two parameters, $\Omega_m$ and $\sigma_8$ (which are qualitatively different for cosmologists). We find that voids have no difference in weights for predicting $\Omega_m$, with average of $55.4\%$ weight for $\Omega_m$ and $\sigma_8$, and $54.0\%$ respectively. Clusters, especially high-significance ones, have higher weights for predicting $\sigma_8$, with average of $14.8\%$ weight for $\sigma_8$ over $8.8\%$ weight for $\Omega_m$. With relaxed thresholds of ($\geq +2\sigma$) for clusters ($\leq 0$) for voids, the whole distribution of weights can be seen from the histograms in Figure 5.

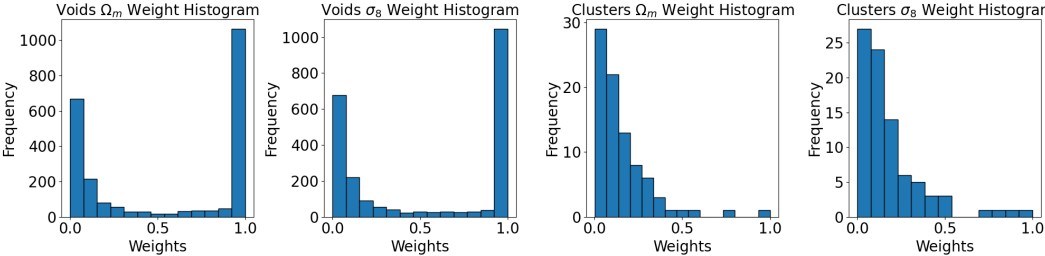

Figure 5: Voids (left two) are used more (have higher weights) in prediction, weighing 100% in about half the cases. Clusters (right two) are used less (have lower weights) in general, but more in predicting $\sigma_8$ than $\Omega_m$.

2. Using a higher threshold of $+2$ or $+3\sigma$ gives the clusters higher weight especially for $\sigma_8$ than with a lower threshold of $+1\sigma$. This aligns with the cosmology concept that rarer clusters with high standard deviation are more sensitive to $\sigma_8$, the parameter for fluctuations.

3. In general, the voids have higher weights for prediction than the clusters in the noiseless maps we have used. This is consistent with previous work (Matilla et al., 2020) that voids are the most important feature in prediction. This finding was intriguing to cosmologists. Given that the previous work relied on gradient-based saliency maps, it is important that we find consistent results with our attention-based wrapper.

It will be interesting to explore how these results change as we mimic realistic data by adding noise and measurement artifacts. Other aspects worth exploring are the role of "super-clusters" that contain multiple clusters, and how to account for the fact that voids occupy much larger areas on the sky than clusters (i.e., should we be surprised that they perform better?).

## 6 RELATED WORKS

**Post-hoc Attributions.** There have been a lot of previous work in attributing deep models post-hoc. One way is to use gradients of machine learning models, including using gradients themselves (Selvaraju et al., 2016; Baehrens et al., 2009; Simonyan et al., 2014b; Bastings & Filippova, 2020), gradient $\times$ inputs (Sundararajan et al., 2017a; Denil et al., 2014; Smilkov et al., 2017) and through propagation methods (Ribeiro et al., 2018; Springenberg et al., 2014; Bach et al., 2015; Shrikumar et al., 2017; Montavon et al., 2017).

Another type of attribution includes creating a surrogate model to approximate the original model (Ribeiro et al., 2016; Lundberg & Lee, 2017; Laugel et al., 2018). Other works use input perturbation including erasing partial inputs (Petsiuk et al., 2018; Vasu & Long, 2020b; Kaushik et al., 2020; Li et al., 2017; Kádár et al., 2017; Ribeiro et al., 2018; De Cao et al., 2020) and counterfactual perturbation that can be manual (Kaushik et al., 2020) or automatic (Calderon et al., 2022; Zmigrod et al., 2019; Amini et al., 2022; Wu et al., 2021). While the above methods focus on individual features, Tsang et al. (2020a) investigates feature interactions. Multiple works have shown the failures of feature attributions (Bilodeau et al., 2022; Sundararajan et al., 2017b; Adebayo et al., 2018; Kindermans et al., 2019).

**Built-in Attributions.** For built-in feature attributions, one line of work first predict which input features to use, and then predict using only the selected features, including FRESH (Jain et al., 2020) and (Glockner et al., 2020). FRESH (Jain et al., 2020) has a similar structure as our model, with a rationale extractor that extracts partial input features, and another prediction model to predict only on the selected features. The difference is that FRESH only selects one group of features while we select multiple and allow different attribution scores for each group. Another line of work learns different modules when using different input features, including CAM (Lou et al., 2012), GA$^2$M (Lou et al., 2013), and NAM (Agarwal et al., 2021). The key difference of our work from these works is that we use grouped attributions to model complex groups of features with different sizes, while previous works attribute to input features individually or in pairs. Other works have explored building feature attributions with guarantees of stability (Xue et al., 2023) and minimality (Bassan & Katz, 2023). Our

work has a similar faithfulness guarantee to Lyu et al. (2023), which studies faithful chain-of-thought explanations.

**Grouped Attributions.**   Previous works have explored interactions between features. Archipelago (Tsang et al., 2020a) finds pairwise interactions between features by perturbing both in and then merging pairs to create larger groups. Integrated directional gradients (Sikdar et al., 2021) computes integrated gradients and groups found by parsers. Parallel Local Search (PLS) (Hase et al., 2021) starts with a random group and then adds randomly selected features based on an objective to create groups. FRESH (Jain et al., 2020) uses the attention mechanism in the Transformer model to select one group of features for built-in attributions.

## 7   CONCLUSION

In this paper, we identify a fundamental barrier for feature attributions in satisfying faithfulness tests. These limitations can be overcome when using grouped attributions which assign scores to groups of features instead of individual features. To generate faithful grouped attributions, we develop the SOP model, which uses a group generator to create groups of features, and a group selector to score groups and make a faithful prediction. The group attributions from SOP improve upon standard feature attributions on the majority of insertion and deletion-based interpretability metrics.

We applied the faithful grouped attributions from SOP to discover cosmological knowledge about the growth of structure in an expanding universe. Our groups are semantically meaningful to cosmologists and revealed new properties in cosmological structures such as voids and clusters that merit further investigation. We hope that this work paves the way for further scientific discoveries from faithful explanations of deep learning models that capture complex and unknown patterns.

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

## A    RELATED WORKS COMPARISONS

A comparison of the properties of different feature attribution methods can be found here.

| Name | Faithful | Grouped | Model-agnostic |
|------|----------|---------|----------------|
| LIME (Ribeiro et al., 2016) | No | No | Yes |
| SHAP (Lundberg & Lee, 2017) | No | No | Yes |
| RISE (Petsiuk et al., 2018) | No | No | Yes |
| GradCAM (Selvaraju et al., 2016) | No | No | Yes |
| IntGrad (Sundararajan et al., 2017a) | No | No | Yes |
| NAM (Agarwal et al., 2021) (Neural Additive Models) | Yes | No | Yes |
| Archipelago (Tsang et al., 2020b) | No | Yes | Yes |
| FRESH (Jain et al., 2020) | Yes | Yes | No |
| IDG (Sikdar et al., 2021) (Integrated Directional Gradients) | No | Yes | No |
| PLS (Hase et al., 2021) (Parallel Local Search) | No | Yes | Yes |
| SOP (ours) | Yes | Yes | Yes |

Table 2: Compare properties of different existing feature attribution methods

## B    THEOREM PROOFS FOR SECTION 2

**Theorem 1** (Deletion Error for Monomials). *Let $p : \{0,1\}^d \to \{0,1\}$ be a multilinear monomial function of $d \leq 20$ variables, $p(x) = \prod_{i=1}^{d} x_i$. Then, there exists an $x$ such that any feature attribution for $p$ at $x$ will incur an approximate lower bound of $e^{\gamma_1 d + \gamma_0}$ total deletion error, where $(\gamma_1, \gamma_0) = (0.664, -1.159)$.*

*Proof.* Let $x = \mathbf{1}_d$, and let $\alpha \in \mathbb{R}^d$ be any feature attribution. Consider the set of all possible perturbations to the input, or the power set of all features $\mathcal{P}$, We can write the error of the attribution under a given perturbation $S \in \mathcal{P}$ as

$$\text{error}(\alpha, S) = \left| 1 - \sum_{i \in S} \alpha_i \right| \tag{5}$$

This captures the faithfulness notion that $\alpha_i$ is faithful if it reflects a contribution of $\alpha_i$ to the prediction. Then, the feature attribution $\alpha^*$ that achieves the lowest possible faithfulness error over all possible subsets is

$$\alpha^* = \arg\min_{\alpha} \sum_{S \in \mathcal{P}} \text{error}(\alpha, S) \tag{6}$$

This can be more compactly written as

$$\alpha^* = \arg\min_{\alpha} \mathbf{1}^\top |\mathbf{1} - M\alpha| \tag{7}$$

where $M_{ij} = \begin{cases} 1 & \text{if } j \in S_i \\ 0 & \text{otherwise} \end{cases}$ for an enumeration of all elements $S_i \in \mathcal{P}$. This is a convex program that can be solved with linear programming solvers such as CVXPY. We solve for $\alpha^*$ using ECOS in the `cvxpy` library for $d \in \{2, \ldots, 20\}$. To fit the exponential function, we fit a linear model to the log transform of the output which has high degree of fit (with a relative absolute error of 0.008), with the resulting exponential function shown in Figure 2a. □

**Lemma 1** (Insertion Error for Monomials). *Let $p : \{0,1\}^d \to \{0,1\}$ be a multilinear monomial function of $d$ variables, $p(x) = \prod_{i=1}^{d} x_i$. Then, for all $x$, there exists a feature attribution for $p$ at $x$ that incurs at most 1 total insertion error.*

*Proof.* Consider $\alpha = 0_d$. If $x \neq 1_d$ then this achieves 0 insertion error. Otherwise, suppose $x = 1_d$. Then, for all subsets $S \neq \{1, \ldots, d\}$, $p(x_S) = 0 = \sum_{i \in S} \alpha_i$ so $\alpha$ incurs no insertion error for all but one subset. For the last subset $S = \{1, \ldots, d\}$, the insertion error is 1. Therefore, the total insertion error is at most 1 for $\alpha = 0_d$. $\square$

**Theorem 2** (Insertion Error for Binomials). *Let $p : \{0,1\}^d \to \{0,1,2\}$ be a multilinear binomial polynomial function of $d$ variables. Furthermore suppose that the features can be partitioned into $(S_1, S_2, S_3)$ of equal sizes where $p(x) = \prod_{i \in S_1 \cup S_2} x_i + \prod_{j \in S_2 \cup S_3} x_j$. Then, there exists an $x$ such that any feature attribution for $p$ at $x$ will incur an approximate lower bound of $\exp(\lambda_2 d + \lambda_1) + \lambda_0$ error in insertion-based faithfulness tests, where $(\lambda_2, \lambda_1, \lambda_0) = (0.198, 1.332, 4.778)$ and $d \leq 20$.*

*Proof.* Consider $x = 1_d$. The addition error for a binomial function can be written as

$$\text{error}(\alpha, S) = \left| \sum_{i \in S} \alpha_i - 1[S_1 \cup S_2 \subseteq S] - 1[S_2 \cup S_3 \subseteq S] \right| = |M_S^\top \alpha - c_S| \tag{8}$$

where $(M_S, c_S)$ are defined as $(M_S)_i = \begin{cases} 1 & \text{if } i \in S \\ 0 & \text{otherwise,} \end{cases}$ and $c_S$ contains the remaining constant terms. Then, the least possible insertion error that any attribution can achieve is

$$\alpha^* = \arg\min_\alpha \sum_{S \in \mathcal{P}} \text{error}(\alpha, S) = \arg\min_\alpha \mathbf{1}^\top |c - M\alpha| \tag{9}$$

where $(M, c)$ are constructed by stacking $(M_S, c_S)$ for some enumeration of $S \in \mathcal{P}$. This is a convex program that can be solved with linear programming solvers such as CVXPY. We solve for $\alpha^*$ using ECOS in the `cvxpy` library for $d \in \{2, \ldots, 20\}$. To get the exponential function, we fit a linear model to the log transform of the output, doing a grid search over the auxiliary bias term. The resulting function has a high degree of fit (with a relative absolute error of 0.106), with the resulting exponential function shown in Figure 2b. $\square$

**Insertion and Deletion Error for Grouped Attribution.** We can define analogous notions of insertion and deletion error when given a grouped attribution. It is similar to the original definition, however a group only contributes its score to the attribution if all members of the group are present.

**Definition 4.** *(Grouped deletion error) The* grouped deletion error *of a grouped attribution $\beta = \{(S_i, c_i)\}_{i=1}^{G}$ for a model $f : \mathbb{R}^d \to \mathbb{R}$ when deleting a subset of features $S$ from an input $x$ is*

$$\text{GroupDelErr}(\alpha, S) = \left| f(x) - f(x_{\neg S}) - \sum_{i : S \subseteq S_i} c_i \right| \tag{10}$$

**Definition 5.** *(Grouped insertion error) The* grouped insertion error *of an feature attribution $\beta = \{(S_i, c_i)\}_{i=1}^{G}$ for a model $f : \mathbb{R}^d \to \mathbb{R}$ when inserting a subset of features $S$ from an input $x$ is*

$$\text{GroupInsErr}(\alpha, S) = \left| f(x_S) - f(0_d) - \sum_{i : S \subseteq S_i} c_i \right| \tag{11}$$

**Corollary 1.** *Consider $p_1$ and $p_2$, the polynomials from Theorem 1 and Theorem 2. Then, there exists a grouped attribution with zero deletion and insertion error for both polynomials.*

*Proof.* Let $[d]$ denote $\{1, \ldots, d\}$. First let $p_1(x) = \prod_i x_i$ and consider a grouped attribution with one group, $\beta = \{([d], 1)\}$. Then, no matter what subset $S$ is being tested, $S \subset [d]$ is always true, thus:

$$\text{GroupDelErr}(\beta, S) = \left| f(x) - f(x_{\neg S}) - \sum_{i : S \subseteq m_i} s_i \right| = |1 - 0 - 1| = 0$$

Next let $p_2(x) = \prod_{i \in S_1 \cup S_2} x_i + \prod_{j \in S_2 \cup S_3} x_j$ and consider a grouped attribution with two groups, $\beta = \{(S_1 \cup S_2, 1), (S_2 \cup S_3, 1)\}$. If $S = [d]$, then

$$\text{GroupInsErr}(\beta, S) = \left| f(x_S) - f(0) - \sum_{i: S \subseteq S_i} c_i \right| = 2 - 0 - (1 + 1) = 0$$

If $S$ empty, then the insertion error is trivially 0. Otherwise suppose $S$ is missing an element from one of $S_1$ or $S_3$. WLOG suppose it is from $S_1$ but not $S_2$ or $S_3$. Then,

$$\text{GroupInsErr}(\beta, S) = \left| f(x_S) - f(0) - \sum_{i: S \subseteq S_i} c_i \right| = 2 - 1 - (1) = 0$$

Otherwise, suppose we are missing elements from both $S_1$ and $S_3$. Then,

$$\text{GroupInsErr}(\beta, S) = \left| f(x_S) - f(0) - \sum_{i: S \subseteq S_i} c_i \right| = 2 - 0 - (1 + 1) = 0$$

Lastly, suppose we are missing elements from $S_2$. Then,

$$\text{GroupInsErr}(\beta, S) = \left| f(x_S) - f(0) - \sum_{i: S \subseteq S_i} c_i \right| = 0 - 0 = 0$$

Thus by exhaustly checking all cases, $p_2$ has zero grouped insertion error. $\square$

### B.1 DISCUSSION ON THEORETICAL PROOFS

While our theorems and that of Bilodeau et al. (2022) both present impossibility results for feature attributions, we kindly point out that the posed characterization of our results is incorrect, both on the assumptions and the resulting theorem.

Bilodeau et al. (2022) put forth a result that says (put simply) that linear models cannot accurately capture complex models, where complexity is measured by having a large number of piece-wise linear components. Indeed, if we had shown that a linear model is not a good approximation of a highly non-linear model, then this would not be a novel contribution. This is also an unsurprising result (it is not surprising that a linear model cannot approximate a *highly non-linear* model).

However, our result paints a significantly bleaker picture: we show that a linear feature attribution is unable to model the extremely *simple* functions in our theorems. Our examples distill the problem to the fundamental issue in its purest form: correlated features. Specifically, we show feature attribution is impossible with only *one* group of correlated features. This is the **polar opposite** assumption than that of Bilodeau et al. (2022), and we argue that it is more surprising for feature attribution to be impossible for simpler functions than for complex functions.

Second, we provide not only a negative impossibility result for standard feature attributions, but also a positive result for grouped feature attributions that provides a path forward and motivates the approach in our submission. This is in contrast to Bilodeau et al. (2022), which only presents negative impossibility results in standard feature attributions without clear suggestions on where to go.

In summary, our theoretical results differ in Assumption (we assume simple functions with a single correlation whereas Bilodeau et al. (2022) assume complex functions with many piece-wise linearities) Theoretical results (we show both positive and negative results, whereas Bilodeau et al. (2022) only show negative results) With this extended discussion, we hope the reviewer can better understand the theoretical results from both our paper and from Bilodeau et al. (2022).

## C ALGORITHM DETAILS

Our algorithm is formalized in Table 1. For GroupGen, we are using weights from all the queries to keys with multiple heads without reduction. Therefore we have $d$ number of group per attention head, and a total of $d \times d_a$ groups for $d_a$ attention heads.

---

**Algorithm 1** Sum-of-Parts Models

---

**Require:** Group Generator GroupGen, Group Selector GroupSelect
**Require:** Input Features $X$, Prediction Model $f$

$\quad S_1, S_2 \ldots, S_G \leftarrow$ GroupGen$(X)$ $\hfill \triangleright$ Group Generation

$\quad$**for** $j = 1 \ldots G$ **do**
$\quad\quad z_i \leftarrow f(S_i \odot X)$ $\hfill \triangleright$ Embedding Grouped Input Features
$\quad$**end for**

$\quad (c_1, y_1), \ldots, (c_G, y_G) \leftarrow$ GroupSelect$(z_1, \ldots, z_G)$ $\hfill \triangleright$ Group Selection

$\quad y \leftarrow \sum_i^G c_i \cdot y_i$ $\hfill \triangleright$ Sum-of-Parts

---

For GroupGen, we first use either a patch segmenter (for ImageNet) or a contour segmenter (for Cosmogrid) to divide the image into segments. For patches, we separate them into 16x16 patches, and use a simple convolutional layer with patch size 16 and stride size 16 to convert each patch into an embedding of the patch. For contours, we use watershed segmenter in Scikit-learn, and use the pooler output (last hidden state of the [CLS] token) of each segment (masking out everywhere that is not the segment) as the embedding of the segment.

Then, we use the multiheaded self-attention described in the paper to assign scores to each patch. For all the segments, we have attention to all other segments. For example, for an 224x224 image in ImageNet, we have 196 patches, so when we use 2 heads in the multiheaded attention, we have potentially 196x2 group candidates. In order to reduce computation, we randomly dropout others and select only 20 groups to use during training, and also cap it at 50 for testing.

We use sparsemax which projects softmax scores to a simplex so that some scores are zeroed out completely. We use this so that each group does not contain all the segments in the image.

## D    EXPERIMENT DETAILS

### D.1    TRAINING

**ImageNet**    ImageNet (Russakovsky et al., 2015) contains 1000 classes for common objects. We use a subset of the first 10 classes for our evaluation. We use a finetuned vision transformer model from HuggingFace [2] for ImageNet.

### D.2    EVALUATION DETAILS

#### D.2.1    ACCURACY

We evaluate on accuracy to measure if the wrapped model has comparable performance with the original model, following Jain et al. (2020). For post-hoc explanations, the performance shown will be the performance of the original model, since they are not modifying the model.

#### D.2.2    DELETION AND INSERTION

Petsiuk et al. (2018) proposes insertion and deletion for evaluating feature attributions for images.

**Deletion**    Deletion deletes groups of pixels from the complete image at a time, also starting from the most salient pixels from the attribution. If the top attribution scores reflect the most attributed features, then the prediction consistency should drop down from the stsart and result in a lower deletion score.

**Insertion**    Insertion adds groups of pixels to a blank or blurred image at a time, starting with the pixels deemed most important by the attribution, and computes the AUC of probability difference between predictions from the perturbed input and original input. If the top attribution scores faithfully

---

[2]https://huggingface.co/google/vit-base-patch16-224

reflect the most attributed features, then the prediction consistency should go up from the start and result in a higher insertion score.

**Grouped Insertion and Deletion** For a standard attribution, it orders the features. Each feature is a group, and thus we test by deleting or inserting in that order. For a grouped attribution, the natural generalization is then to delete or insert each group at each time. Besides the regular version of insertion and deletion, we also use a grouped version. For deletion, instead of removing a fixed number of pixels every step, we delete a group of features. If the features to remove overlaps with already deleted features, we only remove what has not been removed. The same is performed for grouped insertion when adding features. To get the groups, we use groups generated from SOP.

### D.2.3 Sparsity

Having sparse explanations helps with interpretability for humans. We evaluate the sparsity of our grouped attributions by count the number of input features in each group $i$, and then average the count for all groups with non-zero group score $c_i$.

$$\# \text{ group nonzeros} = \frac{\sum_i (|S_i| \mathbb{1}(c_i \geq 0))}{|X_i|}$$

The fewer number of nonzeros implies more sparsity, and thus better human interpretability. On ImageNet, we get around 60% nonzeros. This shows that SOP produces groups that are sparse.

### D.3 Examples

Examples of previous attribution methods and our method can be found in Table 3 and Table 4 respectively.

Examples for SOP insertion/deletion and grouped insertion/deletion can be found in Figure 6.

## E Case Study: Cosmology

In our collaboration with cosmologists, we identified two cosmological structures learned in our group attributions: voids and clusters. In this section, we describe how we extracted void and cluster labels from the group attributions.

Let $S$ be a group from SOP when making predictions for an input $x$. Previous work (Matilla et al., 2020) defined a cluster as a region with a mean intensity of greater than $+3\sigma$, where $\sigma$ is the standard deviation of the intensity for each weak lensing map. This provides a natural threshold for our groups: we can identify groups containing clusters as those whose features have a mean intensity of $+3\sigma$. Specifically, we calculate

$$\text{Intensity}(x, S) = \frac{1}{|S|} \sum_{i:S_i>0} x_i$$

Then, a group $S$ is labeled as a cluster if $\text{Intensity}(x, S) \geq 3\sigma$. Similarly, Matilla et al. (2020) define a void as a region with mean intensity less than 0. Then, a group $S$ is labeled as a cluster if $\text{Intensity}(x, S) < 0$.

### E.1 Cosmogrid Dataset

CosmoGridV1 is a suite of cosmological N-body simulations, spanning different cosmological parameters (including the parameters $\Omega_m$ and $\sigma_8$ considered in this work). They have been produced using a high performance N-body treecode for self-gravitating astrophysical simulations (PKDGRAV3). The output of the simulations are a series of snapshots representing the distribution of matter particles as a function of position on the sky; each snapshot represents the output of the simulation at a different cosmic time (and, therefore, represents a snapshot of the Universe at a different distance from the observer). The output of the simulations have been post-processed to produce weak lensing mass maps, which are weighted and projected maps of the mass distribution and that can be estimated from current weak lensing observations (e.g., Jeffrey et al. (2021)).

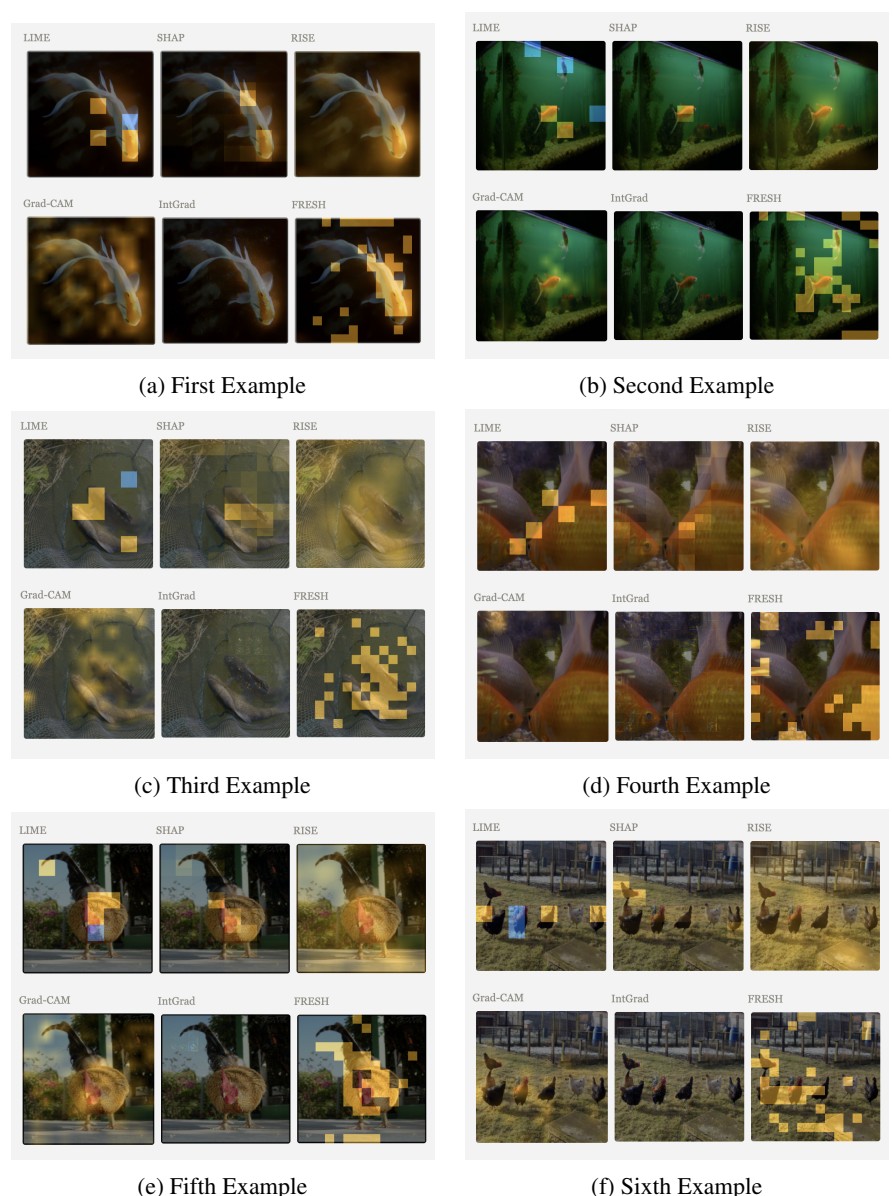

Table 3: Example attributions from previous methods. The overlaying on top of images shows the feature attribution scores of each attribution method. The orange overlay indicates high positive importance from the method for predicting the class, and the blue overlay indicates negative importance.

### E.2 PREPROCESSING

For input features used in CosmoGridV1, we segment the weak lensing maps using a contour-based segmentation method watershed (Beucher, 2023) implemented in scikit-image. We use watershed instead of a patch segmenter because watershed is able to segment out potential input features that can constitute voids and clusters. In our preliminary experiments, we also experimented with patch, quickshift (Grady, 2006) for segmentation. Only the model finetuned on watershed segments is able to obtain comparable MSE loss as the original model.

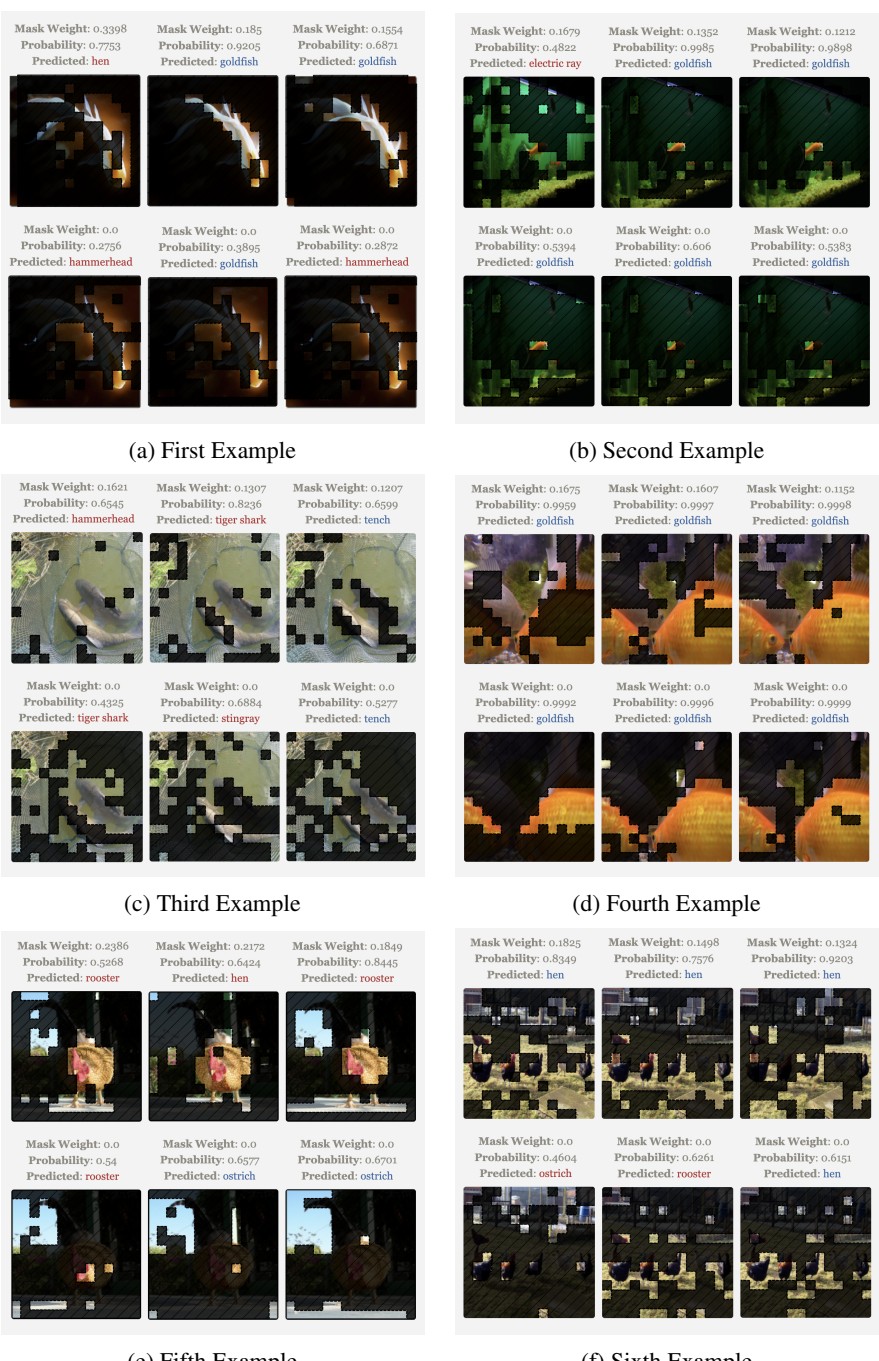

Table 4: Exampled from Grouped attributions from SOP. The masked-out areas in the images are zeroed out, and the unmasked areas are preserved features for each group. The first row shows the groups that are weighted most in prediction. The second row shows groups that are weighted the least (0) in prediction. The probability for each group's predicted class is shown. Predicted classes marked blue are what is consistent with the final aggregated prediction, while red are inconsistent.

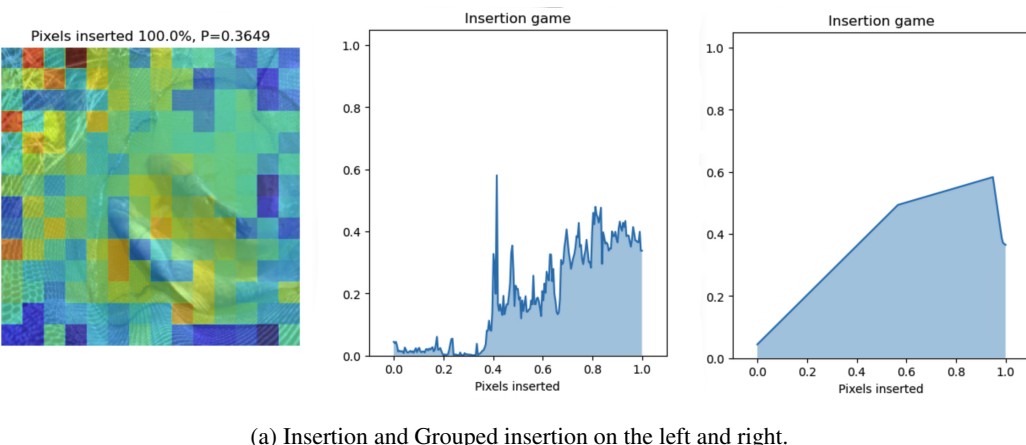

(a) Insertion and Grouped insertion on the left and right.

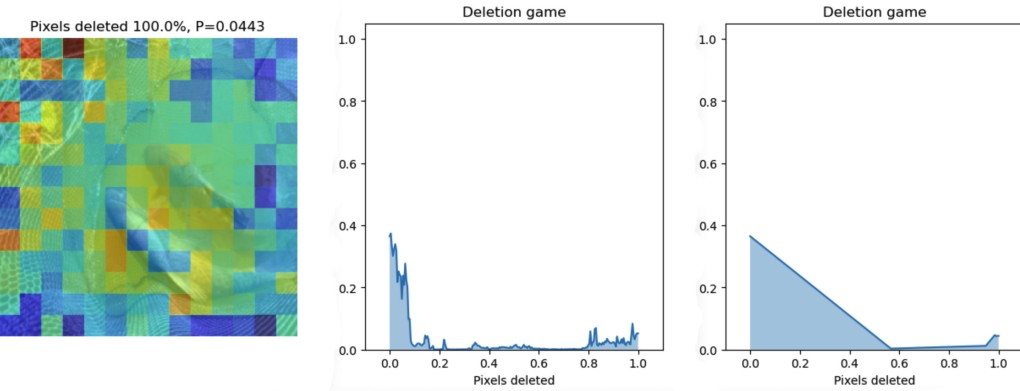

(b) Deletion and Grouped deletion on the left and right.

Figure 6: Insertion and deletion in the standard and grouped forms.

