# OpenReview forum: "Sum-of-Parts Models: Faithful Attributions for Groups of Features"
_ICLR.cc/2024/Conference — Submitted to ICLR 2024_

### Official Review · Reviewer_tAA6 · 2023-10-30

**Soundness:** 3 good
**Presentation:** 1 poor
**Contribution:** 2 fair
**Rating:** 5
**Confidence:** 3

**Summary:**

The authors propose an attribution method — SOP — which is based on groupings of features rather than individual features. They first motivate SOP by demonstrating how attributions based on individual features have intrinsic shortcomings. They then described the SOP and showed its superiority over a few popular, existing methods. Finally, they described a case study of using SOP as an XAI method with expert cosmologists!

**Strengths:**

- The authors presented some theory to motivate their method.
- The method presented is technically sound.
- The real-world use case is worth applauding.

**Weaknesses:**

(These are mostly about clarity.)
- The work’s level of novelty can be made much clearer if the authors could spell out the similarity and difference between SOP and some existing methods.
- While the authors explain the high-level rationales of the method and evaluation, I feel that descriptions of some important details are either missing or too terse.
- For the case study, it is unclear to me if the insights were more from SOP or simply from combining a priori knowledge and the outcomes of the model to be explained.

**Questions:**

**On relation to existing methods:**

In general, I feel many methods share similar features with SOP’s Group Generator and Group Selector. Thus, it would be good to highlight the similarity and difference so that the reader can better judge the novelty of the approach. In particular, could the authors highlight in more detail how their method is similar and different to RISE and LIME. The masks in RISE seem to be equivalent to the groups in SOP, and RISE also combines the model’s prediction z’s (see Equations 1-6 in the RISE ref). LIME learns the weights on masked areas (see Section 3.4 in the LIME ref). Both methods seem to have some form of Group Generator and Group Selector. I think spelling out the similarities and differences would help a lot.

**On model and evaluation details:**

Reading through Sections 2—4, I gathered the following questions related to the implementation and evaluation of the model. Overall, I feel some important details are missing or unclear in the paper.

Questions on Section 2:
- The powerset P is not used in the equations. Should the last line of page 2 read \Sum{S \in P} instead of just S?
- In Definition 1, I don’t quite understand why the difference is between \Delta f and \sum a. A delta (difference) and a sum seem to be very different quantities to compare. Same question for Definition 2.
- Why is the range of \alpha_i and f(x) R and not [0,1]?
- How is the y-axis in Figure 2 calculated? The scale is not what I expected. I was expecting something between 0 and 1, assuming f(x) gives a probability.
- Is the exponential increase in Figure 2 simply a consequence of summing over a powerset of differences (and the size of the powerset grows exponentially with d)?
- The paper mentions that a linear model achieves 0 error as defined. Is this metric only 0 for linear models? Can a non-linear model have 0 error? Does SOP have 0 error? Perhaps going through the calculation of a simple case would help.
- Could the authors clarify in what sense is SOP more faithful than LIME, RISE, and GradCAM? Perhaps theorems 1 and 2 give the answers already, but I am still not very clear after finishing reading the paper.

Questions on Section 3:
- How are the parameters learned? What’s the objective function and the learning algorithm?
- How does the Group Generator affect the outcome? For example, does the method work less well if a random generator is used? How about a generator like that used in Bayesian Teaching saliency map (see Methods section in [1])?
- Want more clarity on the definition, construction, and intuition of the W’s, C’s, the sparsemax.
- What does the “, C z^T” (especially the comma) mean in Equation (4)?

[1] Yang, Scott Cheng-Hsin, et al. "Mitigating belief projection in explainable artificial intelligence via Bayesian teaching." Scientific reports 11.1 (2021): 9863.

Questions on Section 4:
- For the insertion and deletion metric, how did the authors choose the features to be removed one at a time from the SOP? Is some kind of averaging over groups done first?
- Similarly, for the group insertion and deletion metric, how did the authors make the groups from the baseline methods? I read Appendix C.2 and can guess what might have been done, but I think it can be described more precisely. In general, it would be good to describe the implementations of the metrics more precisely.
- It will be good to show the curves for the insertion/deletion metric and the grouped versions.

**On the case study:**

It seems to me that the Void and Cluster types are known things to begin with, so it’s unclear how the SOP groups helped provide insight. Also, what is the advantage of looking at the scores/weights c of the Group Selector compared to just using the backbone model’s output z as the scores? In other words, I am still not very clear how the Group Generator offered more insight than the a priori known Void and Cluster, and how the Group Selector offered more insights than just the backbone model’s prediction.

In general, while the SOP may have smaller group deletion and insertion error, it is more complex than a single attribution map. SOP is really also a ML model. This raises the question of how the user should use the SOP. Although the case study offers some indirect hints, it is not very clear to me how I should go about using it.

---

> ### Author Response · Authors · 2023-11-23
>
> We thank the reviewer for the constructive review.
>
> ### W1.
> We have added a table to show the similarity and differences between SOP and other methods in Appendix A Table 2.
>
> ### W2.
> Thank you for the suggestion of writing. We include more details for the method in Appendix C in the updated manuscript.
>
> ### W3.
> It is true that we can combine the a priori knowledge with the outcomes of any explainable model. However, the interpretation of different explanations are different. Our model gives faithful groups and their contributions to the final prediction, and the case study shows that the faithful groups correspond to the a priori knowledge, meaning that the semantically meaningful groups are contributing to the prediction.
>
> ### Q1.
> Although RISE is also combining the masks, their prediction is not based on the outputs from the masks. So the post-hoc masks are not directly contributing to the prediction, where SOP's masks are directly used in making the prediction.
>
> ### Q2.
> #### Section 2:
> 1. Yes. We apologize for the confusion. It should be $\sum{S \in P}$.
> 2. $\Delta f$ is the difference between predicting with the whole input x and x without subset $S$. \sum a is the sum of attribution scores for features in S. We would hope that attribution scores for each feature faithfully represent how much that feature contributes to the prediction of $f(x)$. Thus $f(x) - f(x_{\lnot S})$ should be similar to \sum a.
> 3. In the general case, they could have a real value contribution to the prediction.
> 4. The total deletion error is $\sum_{S\in\mathcal{P}} \mathrm{DelErr}(\alpha,S)$ where $\mathcal P$ is the powerset of $\{1,\dots, d\}$, as in Definition 1. Thus y-axis is the sum over the whole powerset.
> 5. Yes.
> 6. By definition here, only a linear model can achieve 0 error. SOP have 0 error. For SOP, we have access to the internal process of the new model, which is the linear combination of different passes of the old model. As a simple example, suppose $p(x) = x1 * x2 + x2 * x3$ where $x1, x2, x3 \in \{0, 1\}$, and SOP selects group $S1 = {x1, x2}$ and $S2 = {x2, x3}$, and assigns score $\alpha_1 = 1$, $\alpha_2 = 1$. Then, when $x1, x2, x3=1$, $InsErr = p(x_S) - p(0_d) - \sum_i \alpha_i = 2 - 0 - (1 + 1) = 0$. And if we only select $S1$, then $x1 = 1, x2 = 1, x3 = 0$, $InsErr = 1 - 0 - 1 = 0$. This is true for all cases. For images, this is also the same.
> 7. SOP first generates the groups, and aggregates the separately obtained results from the groups. In this way, the prediction from each group is inherently based only on the group and different from post-hoc methods such as LIME, RISE, and GradCAM. Also the groups in SOP overcome the inherent limitation of individual feature attributions.
>
> #### Section 3:
> 1. The extra attention layers in GroupGen and GroupSelect are learned end-to-end and the model is trained on cross-entropy for classification.
> 2. We have not experimented with other Group Generators. It is true that our group generator is not the best one yet. We leave exploring different group generators for future work.
> 3. The W’s are learned end-to-end as part of the attention mechanism. The C is taken from the trained model’s classification layer, as it is the representation of the classes. The intuition behind this is to compute which group is most similar to each class and should be used for predicting that class. Sparsemax is used to completely zero-out some segments and some groups for a more human-interpretable set of groups.
> 4. $C z^T$ means doing matrix multiplication of the class weight matrix and the pooler output of each group, which is equivalent to $y_1, … y_G$. The comma means that there are two things outputted from GroupSelect. The first is the weights for each group for each class $c_1, …, c_G$. The second is the prediction from each group $y_1, …, y_G$.
>
> #### Section 4:
> 1. We aggregate the grouped attributions with their group weights. Although this is not how our groups are intended to be used, we need to aggregate to be able to compare with other standard feature attribution methods.
> 2. For group insertion and deletion metrics, we use the same groups generated from SOP for evaluating other baseline, as they do not inherently come with groups.
> 3. We have added insertion and deletion examples for grouped and standard cases in Appendix D.3.
>
> #### Case study
> 1. While voids and clusters are known, it has been unclear how much they contribute to omega and sigma for prediction. The research question here is how much they affect omega and sigma, and the weights from Group Selector offer some insights for that. Just using the backbone model’s output z is not faithful explanation of how this group contributes to the prediction.
> 2. One way to use SOP would be to look at top k groups affecting the prediction. We can also plot the histogram of different clusters of groups of groups that correspond to different a priori knowledge concept (such as voids and clusters) to obtain global explanations.

---

### Official Review · Reviewer_uA3B · 2023-10-30

**Soundness:** 2 fair
**Presentation:** 3 good
**Contribution:** 3 good
**Rating:** 5
**Confidence:** 4

**Summary:**

This paper proposes a method for identifying groups of features (instead of individual features) which are important for a model’s prediction. The method trains an additional auxiliary model to reproduce the original model’s prediction. The auxiliary model learns to generate sparse masks (which subset the input’s features in different ways), and these subsetted features are fed into the original model, and output embeddings from each feature subset (mask) are projected and summed to the target. That way, one can see which masks were given a high weight in the prediction, and those masks contain groups of features which have high importance. The authors provide some theoretical justification for why (in certain specific situations), groups of features offer more faithful interpretations than individual features. The authors benchmark against several other local-interpretation methods, and also provide a real-world example from cosmology.

**Strengths:**

### Good comparison against other methods and use of different datasets

The authors do a good job of showing that their method (SOP) works well to minimize insertion and deletion errors in the face of correlated features using two image datasets. They also show an application of their method in a real-world example of cosmology.

**Weaknesses:**

### There should be more analyses on the quality/interpretability of the feature groups (masks)

Although the authors have done a good job on showing how their method offers improvements in insertion/deletion faithfulness, there is much less shown about the _quality_ of the feature groups/masks which are learned by SOP. The cosmology example is a good singular anecdote, but a more global analysis is needed to show that the interpretability of the feature groups is good. If the feature groups are very noisy and spread out all over an image (for example), then the method would not be very useful compared to another method which gives more contiguous or otherwise interpretable features. This is one of the core pieces that’s holding this paper back, in my opinion, and it would be great to see: 1) some examples of identified features (i.e. like saliency maps) for some examples of images, comparing SOP with other explanation methods; and 2) a global analysis quantifying the quality of the feature maps, comparing SOP with other explanation methods. It would be great to see if SOP is able to generate feature groups which are faithful but also interpretable (or at least, not significantly less interpretable than other methods).

### Theoretical justification is interesting, but it does not seem entirely applicable to motivate the work

The theoretical justification is certainly appreciated, but it does not appear entirely applicable to this work for a few reasons.

Firstly, Theorem 1, Lemma 1, and Theorem 2 attempt to motivate the identification of feature groups by showing that for some specific models (i.e. multilinear monomial or binomial), there exists an $x$ where for any attributions $\alpha$, there is a subset of features which has high error (exponential in the feature dimension). Importantly, these proofs always assume a fixed $x = \mathbb{1}$. This is just one value of $x$, and even if interpretability is poor for this one value of $x$, one could claim that other values of $x$ might be better, and so focusing on this single instance of $x$ (which is unlikely to appear in real data) is not very strong.

Additionally, even with some given $x$ ($\mathbb{1}$ or otherwise), it might not be realistic to consider the error of the worst possible subset. Not all subsets of features will be useful in interpretation, yet the error of all subsets is being summed together in the objective of the convex program. It may not be realistic to include all such subsets.

Furthermore, Corollary 1 is proven for the case where $x = \mathbb{1}$, and there is no formal justification for a general $x$ or general $p$. This is not a deal-breaker, but it should certainly be clarified that the theoretical motivation is over a toy example. Further justification should also be offered for why it is believed that this toy example can be generalized to real-world data and models. Otherwise, one could claim that all this theoretical justification is unrealistic and therefore inapplicable to the problem that SOP is trying to solve.

### Experiments are limited to images

There are many other data modalities, and it would be nice to see other things outside of image datasets, which have their own biases, and are much easier than other modalities in certain ways. The real-world example, however, helps reduce this limitation of the paper. However, if images are the only data type being considered, it would be more reasonable to adjust the focus of the paper in writing to clarify that the work is focused on image datasets (although it may be tweaked to be applied to other datasets in future work).

On a related note, the paper assumes that the baseline image is the all-black image (all 0), but this is not a general assumption, especially in non-image datasets. Many of the Shapley-value works for interpretability are motivated (in part) by this observation. I don’t expect it to be difficult to adjust the math and the implementation to account for non-zero baselines.

### Some areas in the writing and presentation could be improved
- Equation 1: it would be good to formally define feature attribution $\alpha$; also, the powerset $\mathcal{P}$ is not used anywhere in the equation
- Typo in Equation 8
- In Definition 3, $m_i$ is never used
- Although SOP is being presented as a model-agnostic method, there is some reliance on the model architecture because of the need for a good embedding layer
- The bolding can be misleading in Table 1; the meaning of the italics should be made clear in the caption, and Del of integrated gradients for ImageNet should be bolded
- I did not understand this sentence until the second read of the paper: “In practice, we can initialize the value weight C to the linear classifier of a pretrained model.”
- More details on the architecture would be appreciated in the supplement

**Questions:**

- In the convex program, what is the optimal $\alpha^{*}$ that is found? Because we are fixing $x = \mathbb{1}$ and $p(x)$, technically we already have a good idea of what the “right” attributions should be, and most configurations of attributions would be highly unreasonable (which general interpretability methods would not identify at all)
- There is certainly a connection to Shapley scores, which have the efficiency property; insertion and deletion error are closely related; having some background on this and explanation would be appreciated
- There may also be a connection to Novello, et. al., 2022 (Making Sense of Dependence: Efficient Black-box Explanations Using Dependence Measure), which identifies feature importances based on masks, as well, although feature groups is not a core focus there
- How does one choose the number of groups G? Once G is selected, are all of the G feature groups identified by SOP meaningful? Is there anything that prevents feature groups from being redundant? How can one select the most meaningful feature groups out of the G?
- More on the problem of redundancy, for a single input example, the same feature can be present in multiple masks/subsets; does this pose an issue for interpretability in this framework?

---

> ### Author Response · Authors · 2023-11-23
>
> We thank the reviewer for the constructive review.
>
>
> W1. We have added more images for the groups generated and their comparisons with other explanations in Appendix D.3.
>
>
> W2. Thank you for the suggestions. The theoretical justification is the motivation for creating grouped attributions. It is true that the example we provide is a toy example. We only show the case for when x=1. In real cases of images, we will have much more complicated interactions between features and thus we do not only have one value of x that is affected by not accounting for interactions between features.
>
>
> W3. We focus on images but our method is generic and should be able to extend to other modalities.
>
>
> Yes, it is simple to implement other image baselines. We focus on the faithful-by-construction framework, while can easily modify to achieve other image baselines.
>
>
> W4. Thank you for the suggestions and we will add formal definition of feature attribution and fix the typos and writings.
>
>
> For the embedding layer used in GroupGen in ImageNet and VOC, for patches, we are training a new simple convolutional layer that is the same stride size as its patch size, so the layer does not depend on the model. For Cosmogrid, because we are using segments instead of patches, we use the hidden state from the last layer of the model. So it contains more information than just a first embedding layer. If the original model already has reasonable performance on the end task, it is likely that the embeddings from the last layer are good. But it is true that there is some dependence on the model's internal hidden states for the embedding.
>
>
> Q1. For the monomial example, $\alpha\*$ is 1 for one group of $S_1 = \{1, …, d\}$ of all the features. For the binomial example, $\alpha\*_1 = 1$ for one group $S_1 \cup S_2$, and $\alpha\*_2 = 1$ for another group $S_2 \cup S_3$.
>
>
> Q2. Shapley values are computed by the number of times including a feature while perturbing all other features that still lead to the same output. They have the property that Shapley values for groups can be computed by summing the Shapley values for individual features. However, it is impractical to compute real Shapley values for images as the space of all features in a 224x224 image with pixel values in 3 channels from 0 to 255 is too large. SHAP computes an approximate by doing Monte Carlo sampling. However, SHAP score is not the same as Shapley values. Also, although summing all the shapley values will lead to the same prediction as the original prediction, this is not the same decision process as the original model and thus not faithful.
>
>
> Q3.Thank you for pointing out another related work. [Novello, et. al., 2022] assign scores to patches based on the kernel embedding of the distribution of each patch and the output y. As you also pointed out, they focus on individual feature importance, whereas we focus on importance of grouped features.
>
>
> Q4.​​ Currently the candidate number of groups G is number of features x number of heads (chosen). The sparse multihead attention then choose a subset of it. It is true that some feature groups are redundant. If the two feature groups are exactly the same, then the predictions and scores are also the same. When we interpret the final results, we combine the groups that are equivalent.
>
>
> It is also possible at test time to add an additional process to remove the redundant groups before going through the backbone model.
>
>
> GroupSelect assigns a score to each group for each class. The feature group that contributes the most to the prediction of a class is the one with the highest score of that class.
>
> Q5. Each mask/subset captures a different interaction within the group. Since the prediction does not depend on other features outside the group, it can still be interpreted as how this group of features contributes (without need for other features). If all the features are selected within a group, then it becomes uninterpretable. The groups generated by our group generator have around 60% non-zero features selected, so it should be interpretable.

---

### Official Review · Reviewer_75mu · 2023-10-31

**Soundness:** 2 fair
**Presentation:** 3 good
**Contribution:** 2 fair
**Rating:** 3
**Confidence:** 5

**Summary:**

This paper proposes a model that aims to operate in an interpretable way on representations extracted by a deep learning model, for the purpose of image classification. The motivation for this model is presented through a few theorems on approximation error of linearized explanations against the underlying functions they seek to approximate, in particular in settings where there are feature interactions. The model itself is a sparse attention-based model on hidden states extracted by a vision model backbone. Experiments are conducted with standard input ablation metrics to compare the “explanations” (automatically generated during the sparse model forward pass) with post-hoc explanations obtained by methods like SHAP. Results on ImageNet and VOC 07 data are favorable to the proposed method, though the margins are slim. Finally, a case study is conducted for deep learning in cosmology, and it is suggested that the method is reliable for uncovering new scientific knowledge.

**Strengths:**

- Important: The focus on feature interactions is valuable. We should be moving on from linearized feature attributions, and the proposed method aims to do that.
- Important: The case study is ambitious and a great downstream use case of the proposed method. We should see more papers include case studies of their XAI method. It appears that the XAI method suggests conclusions that are consistent with theoretically known results in cosmology, and this is evidence in favor of the proposed method.
- Important: The paper is clearly written and the results are all well presented. Individual design choices are generally well-motivated, e.g. limitations with linear feature attributions pointing toward feature attribution, and the reasoning for grouped feature insertion/deletion.

**Weaknesses:**

- Very Important: The experiments should have compared with existing methods that attempt to capture feature interactions, including (1) the cited Tsang et al 2020, (2) prior work on integrated gradients with feature attribution (https://aclanthology.org/2021.acl-long.71/), and (3) non-parametric search methods for insertion/deletion metrics (https://arxiv.org/abs/2106.00786). It has been known for a while that local linear approximations don’t capture feature interactions, and several methods have been proposed for resolving this.
- Very Important: The primary quantitative metrics in the paper are not presented with sufficient detail, and this leads to the results being difficult to interpret. Is a .02 improvement of SOP over SHAP good or bad? It sounds very small. Sometimes the improvements are smaller or SHAP outperforms SOP.
- Important: While I like the case study, the conclusions are somewhat mixed. (1) It is said that the method helps cosmologists see that voids are more important for predicting the omega constant than sigma, because the feature attribution is on average 55.4% for omega vs 54% for sigma. Surely this difference is so slight that we cannot conclude much from it. (2) The method’s explanations align with prior domain-specific knowledge (that’s good). (3) It’s said that explanations agree with conclusions from a 2020 study using gradient-based salience, and that it is “important that we find consistent results with our attention-based wrapper.” Why is it important to agree with a fundamentally flawed method from a previous study?
- Important: The proposed method suffers an accuracy tradeoff with using the original deep learning model. The gains in explainability should be very clear in order to justify this. Since the gains in explainability are not very clear, this is a mark against using the proposed sparse attention-based model.
- Of Some Importance: I just want to note that I do not think the theoretical analysis adds much to the paper in its current state. We have seen strong impossibility theorems for rich model classes in the past year (e.g. the cited Bilodeau et al 2022 paper). A result showing that a linear model is not a good approximatation of a very nonlinear model, especially in high dimensions, is not very novel.

**Questions:**

- Do the experiments use the same amount of compute between different explanation methods? Why or why not?
- Another reason it is difficult to interpret the results is that SOP explanations are supposed to be faithful by construction but the explanations get metric scores of .07 or .39, etc. Why not evaluate faithful-by-construction explanations according to a criterion that gives them a perfect score? Then we could compare methods along a Pareto curve showing accuracy vs interpretability tradeoffs among the methods. (I also suspect the reason for this, however, is that a sparse-max attention-based model on deep learning representations is not really close to being as “inherently interpretable” as e.g. a decision tree on tabular data, and the need for automatic quantitative metrics reflects this.)
- Typo: “where P is the powerset.” P not in the definition
- Typo: “monimial”

---

> ### Author Response · Authors · 2023-11-23
>
> We thank the reviewer for the constructive review.
>
>
> W1.
> Thank you for pointing out the missing methods to compare. We havewill added experiments with Archipelago (Tsang et al 2020) for ImageNet and put in the revised Table 1. Due to time constraint, we have not compared with IDG and PLS as they only have code on text and IDG relies on a parser which is not available for images.
>
>
> W2.
> Our goal is to have guaranteed faithfulness which none of the other approaches have. The purpose of the metrics is to be consistent with the literature, but none of these metrics can perfectly capture faithfulness. The fact that our approach can guarantee faithfulness while matching SHAP performance is a good thing.
>
>
> W3.
> We acknowledge a mistake on our part and thank the reviewer for their careful reading. Indeed we have confirmed with the cosmologists and this was a mistake in the text—the comparison between Omega and Sigma for voids was intended to show that there was no difference (54% to 55%), in contrast to the Omega and Sigma for clusters where there is a significant difference (14.8% to 8.8%). We have updated this in the revised manuscript.
>
>
> The previous gradient-based method does not give guarantee that the weights reflect how much the features contribute to the final prediction. Since gradient saliency methods are known to be unfaithful [Gilmer et al. 2020], this cast doubt into the cosmology community as for whether these results were true.
>
>
> In fact, as we state in the third finding in Section 5, our findings are consistent with previous work's finding using gradient-based saliency maps. This can show that our findings are consistent with the findings of existing feature attributions in weak lensing maps, while we give more guarantees in how the weights faithfully reflect how much the groups are used by the model and validate the previous results.
>
>
> It is important that we can validate their results (so we know that their results are correct), so we know for sure that voids can be more important than clusters for predicting these cosmological parameters. This is not to say that our method has to find the consistent results with fundamentally limited feature attribution methods. It would have been possible that we find opposite results.
>
>
> Adebayo, J., Gilmer, J., Muelly, M., Goodfellow, I., Hardt, M., & Kim, B. (2020). Sanity Checks for Saliency Maps. arXiv. https://arxiv.org/abs/1810.03292
>
>
> W4. While our theorems and that of Bilodeau et al 2022 both present impossibility results for feature attributions, we kindly point out that posed characterization of our results is incorrect, both on the assumptions and the resulting theorem.
>
>
> Bilodeau et al. 2022 put forth a result that says (put simply) that linear models cannot accurately capture complex models, where complexity is measured by having a large number of piece-wise linear components. Indeed, we agree that **if** we had shown that a linear model is not a good approximation of a highly non-linear model, then this would not be a novel contribution. This is also an unsurprising result (it is not surprising that a linear model cannot approximate a *highly non-linear* model).
>
>
> However, our result paints a significantly bleaker picture: we show that a linear feature attribution is unable to model the extremely **simple** functions in our theorems. Our examples distill the problem to the fundamental issue in its purest form: correlated features. Specifically, we show feature attribution is impossible with only **one** group of correlated features. This is the **polar opposite** assumption than that of Bilodeau et al. 2022: our theorem shows that standard feature attributions cannot hope to explain even simple functions if they contain just a single group of correlated features, which is more surprising than being unable to explain complex neural networks functions (and subsumes that case).
>
>
> Second, we provide not only a negative impossibility result for standard feature attributions, but also a positive result for grouped feature attributions that provides a path forward and motivates the approach in our submission. This is in contrast to Bilodeau et al. 2022, which only presents negative impossibility results standard feature attributions  without clear suggestions on where to go.
>
>
> In summary, our theoretical results differ in
> Assumption (we assume simple functions with a single correlation whereas Bilodeau et al. 2022 assume complex functions with many piece-wise linearities)
> Theoretical results (we show both positive and negative results, whereas Bilodeau et al. only show negative results)
> With this extended discussion, we hope the reviewer can better understand the theoretical results from both our paper and from Bilodeau et al. 2022. We have also included this discussion in the revision in Appendix B.1.

---

> > ### Comment · Reviewer_75mu · 2023-11-30
> >
> > Thanks for the response! Some more comments below.
> >
> > > W1. Thank you for pointing out the missing methods to compare. We havewill added experiments with Archipelago (Tsang et al 2020) for ImageNet and put in the revised Table 1.
> >
> > Thanks. The performance gains of the paper's proposed method look minor and mixed to me, relative to existing feature interaction attribution methods, but it is also hard to tell due to the metrics not being intuitive.
> >
> > > W2. Our goal is to have guaranteed faithfulness which none of the other approaches have. The purpose of the metrics is to be consistent with the literature
> >
> > There is no such thing as guaranteed faithfulness, intrinsic interpretability, or faithfulness by design. All explanations of an ML system must be interpreted by a human, and the human's mental model of the ML system can be evaluated for its faithfulness to the ML system. This paper does not do that, but defaults to common insertion/deletion metrics from the literature, along which there is seemingly little change relative to past methods. If this paper were proposing an approach for finding, e.g. sparse decision trees, I would be more accepting of a claim that the models are _definitely_ interpretable and we don't need to run more rigorous eval of the interpretability. But a newly designed neural model on top of an existing neural model doesn't promise much interpretability until it is demonstrated empirically to do so.
> >
> > > In summary, our theoretical results differ ...
> >
> > Thanks for the discussion. I appreciate the differences with Bilodeau et al. I think what would benefit this paper, in a second draft, would be to focus on (1) theoretical analysis of feature interactions and what kinds of functions can be captured by linear _or_ feature interaction attribution methods, (2) a new feature interaction method evaluated robustly across many datasets with automatic and human evaluation metrics, OR (3) a case study in cosmology showing more robustly how a method can generate new scientific knowledge. I think right now the paper tries to do all three things and suffers in its convingness for doing so.
> >
> > ---
> >
> > Following the above discussion, I will maintain my score (3) and confidence (5).

---

### Official Review · Reviewer_xXht · 2023-11-02

**Soundness:** 3 good
**Presentation:** 1 poor
**Contribution:** 4 excellent
**Rating:** 5
**Confidence:** 3

**Summary:**

A new way to combine predictions of an existing model so that feature attribution is easy. The method uses an existing model to predict the outcome over a subset of features and then aggregates the individual outcomes to get the final prediction. Results show that the method does not lead to accuracy loss compared to the existing model.

**Strengths:**

- A fresh idea (as far as I can tell) on designing a new way to form model predictions that are interpretable by design. Rather than post-hoc interpretability.
- Results using a cosmology case study that show the practicality of finding groups of features necessary for the algorithm.
- An analysis of insertion and deletion tests that can be used to evaluate interpretability methods.

**Weaknesses:**

- The writing can be improved a lot. It took me a long time to figure out whether the paper is proposing an interpretation technique or a model fitting technique. A simple sentence in the abstract, "we modify an existing model to become interpretable, by using its predictions over a subset of features and then aggregating them", will be useful to readers.

- I have many questions about the SOP model since the writing is unclear. Listing them in the questions section.

- While inspiring, the cosmology evaluation feels incomplete. Without a comparison control set (where cosmologists used LIME or some other method), it is not clear whether the benefits from SOP are unique. Also, there is a subtle problem in the conclusion of results: the results only show that _one_ predictive model tends to weigh voids higher than clusters. It does not say anything about the true process. It is possible that there is another model with same accuracy that weights voids and clusters differently. So I'm not sure what is the goal of the cosmology case study--is it to explain a ML model, or is it to understand the DGP/true model?

**Questions:**

- It is not clear whether the sum-of-parts model requires any weight updates. Will it be better if aggregation parameter C is learnt? Is the backbone model same as an existing model to be explained, or is it a new model? Is the SOP model trained end-to-end or only the backbone is trained and SOP weightes can be inferred?
- Can you take a concrete example (e.g., ImageNet) and describe to readers how exactly you will create the SoP model?
- Can you recreate Figure 5 using LIME or an existing feature attribution method on a standard non-SOP model? How different are the results?
- Regarding Thm 1 and Thm 2, can you comment on counterfactual explanations? Since they are designed for deletion (but can also be defined for insertion), would they work better for Thm 1 and Thm 2? Would the feature attributions from CF explanations be better than other attributions? (for a discussion on CF explanations versus feature attributions, this paper may be helpful: https://dl.acm.org/doi/10.1145/3461702.3462597)

---

> ### Author Response · Authors · 2023-11-23
>
> We thank the reviewer for the constructive review.
>
>
> W1. Thank you for the advice on writing! We had a similar sentence in the abstract, but have used your suggestion to make this more clear in the updated revision.
> > Specifically, we modify black-box models to have faithful interpretations, by aggregating predictions over varying subsets of features.
>
>
> W3. You bring up a great point: it is indeed possible that there is another model with the same accuracy that weights voids and clusters differently. However, in cosmology, we understand so little about the dynamics of dark matter and density fluctuations that understanding how *any* model (that achieves good performance) is weighting voids and clusters is an interesting problem. If there are multiple models that can use different patterns to achieve the same accuracy, then both of these are useful findings for cosmology. The baseline-study from [Matilla et al. 2020] is what our case study builds off from, which uses gradient saliency maps. However, the critiques of gradient saliency maps and their unfaithfulness [Gilmer et al. 2020, Sundararajan et al. 2017] led some of the cosmology community to have doubts on these initial results. Therefore the goal of our case study is twofold: (a)  to confirm whether these saliency-based results are consistent with faithful explanations (finding #3) (b) to help cosmologists learn more about the cosmos (findings #1 + #2).
>
>
> Matilla, José & Sharma, Manasi & Hsu, Daniel & Haiman, Zoltán. (2020). Interpreting deep learning models for weak lensing. Physical Review D. 102. 10.1103/PhysRevD.102.123506.
>
>
> Adebayo, J., Gilmer, J., Muelly, M., Goodfellow, I., Hardt, M., & Kim, B. (2020). Sanity Checks for Saliency Maps. arXiv. https://arxiv.org/abs/1810.03292
>
>
> Mukund Sundararajan, Ankur Taly, and Qiqi Yan. Axiomatic attribution for deep networks, 2017
>
>
> Q1. The aggregation parameter C is learnt. We initialize it from an existing model but it is still being updated. The backbone model is an existing trained model that is completely frozen and not updated—this is typically a traditional deep learning model that has been trained on the same task. This backbone is used within GroupGen and GroupSelect to create the SOP model. The SOP model has additional trainable parameters (the attention weights in GroupGen and GroupSelect) which are trained end-to-end with the backbone model frozen.
>
>
> Q2. For ImageNet, we take a trained Vision Transformer image classification model from HuggingFace as the backbone model. You can imagine this backbone is then sandwiched between the GroupGen and GroupSelect modules to create an SOP model. Specifically:
> The GroupGen module uses an attention layer to select the groups from the 16x16 patches in an 224x224 image.
> Then, each group is passed separately into the backbone ViT model as the original image with patches not in the group masked out.
> The GroupSelect module then uses an attention layer to select a subset of groups to use to make a prediction. This attention layer uses the last hidden states from these group's ViT model outputs as keys and values, while the query is a learnable parameter initialized with the class weights.  These attention weights are then used to aggregate  the predicted logits from each group for each class.
>
>
> Q3. To evaluate how a structure is weighted in making the prediction, we would need to first have the groups of features and then check if they correspond to voids and clusters. For standard feature attribution methods like LIME, they attribute to individual features, and thus do not have a natural group created along the process. We could potentially do it with thresholded LIME. There is still the issue that since the weights from LIME are not faithfully representing how much the model is using these groups of features, even if we make a figure like Figure 5 using LIME, we do not know how to interpret the results.
>
>
> In fact, as we state in the third finding in Section 5, our findings are consistent with previous work's finding using gradient-based saliency maps. This can show that our findings are consistent with the findings of existing feature attributions in weak lensing maps, while we give more guarantees in how the weights faithfully reflect how much the groups are used by the model and validate the previous results.
>
>
> Q4. Counterfactual explanations do not inherently give a score for different groups of features. The paper by [Mothilal et al. 2021] you pointed out does give a way to obtain feature attributions from counterfactual explanations. The feature attribution score for one feature is averaged over multiple counterfactual explanations. Thus, it is still giving standard individual feature attribution which has limitations over grouped attributions.

---

### Official Review · Reviewer_iUfV · 2023-11-08

**Soundness:** 3 good
**Presentation:** 3 good
**Contribution:** 3 good
**Rating:** 6
**Confidence:** 2

**Summary:**

The authors begin by demonstrating how explanations based on singular feature attributions are not always "faithful" in that the attribution does not actually explain the model's reasoning. The authors propose using groups of features to achieve proper feature attribution. Their proposed technique is SOP (Sum-of-Parts). The authors prove that feature attribution has shortcomings in regards to deletion and insertion errors due to their singular nature and thus feature interaction (which isn't captured). The authors compare their results against four different baselines and provide a case study using their technique to make new discoveries in cosmology.

**Strengths:**

The approach makes intuitive sense to me. It is very strong in that it can be applied to any sort of backbone model. The theoretical results (not my strong suit) are easy enough to follow. The comparison to other approaches and case study are very persuasive.

**Weaknesses:**

I found Table 1 very confusing. Certain numbers are bolded and italicized without any explanation as to why (it isn't bolding the best scores). I wish more attention was given to the Group Generator section.

**Questions:**

1. Please elaborate on the Group Generator portion.

---

> ### Author Response · Authors · 2023-11-23
>
> We thank the reviewer for the constructive review. We want to apologize for the confusion on Table one. There was indeed a mistake with the bolding: the deletion score for integrated gradients should be bolded instead of italicized. This is corrected in the revised version. We originally intended the italicized numbers to indicate post-hoc methods, but acknowledge that in hindsight this was more confusing. We have cleaned this up in the revised version.
>
> Q1: In the revision, we’ve updated the appendix (Appendix C) to contain more details on the group generated. We summarize this information for your convenience.
>
> For Group Generator, we first use either a patch segmenter (for ImageNet and VOC) or a contour segmenter (for Cosmogrid) to divide the image into segments. For patches, we separate them into 16x16 patches, and use a simple convolutional layer with patch size 16 and stride size 16 to convert each patch into an embedding of the patch. For contours, we use watershed segmenter in Scikit-learn, and use the pooler output (last hidden state of the [CLS] token) of each segment (masking out everywhere that is not the segment) as the embedding of the segment.
>
> Then, we use the multiheaded self-attention described in the paper to assign scores to each patch. For all the segments, we have attention to all other segments. For example, for an 224x224 image in ImageNet, we have 196 patches, so when we use 2 heads in the multiheaded attention, we have potentially 196x2 group candidates. In order to reduce computation, we randomly dropout others and select only 20 groups to use during training, and also cap it at 50 for testing.
>
> We use sparsemax which projects softmax scores to a simplex so that some scores are zeroed out completely. We use this so that each group does not contain all the segments in the image.

---

### Meta-Review · Area_Chair_i1nS · 2023-12-06

**Metareview:**

The paper proposes Sum-of-Parts (SOP), a novel class of machine learning models designed to provide faithful explanations of model predictions by producing grouped feature attributions.

pros:
+ The proposed Sum-of-Parts (SOP) model's approach is considered intuitive, offering a strong and applicable solution applicable to various backbone models.

cons:
+ the method has an accuracy tradeoff with the original deep learning model and lacks clear and substantial gains in explainability to justify the sacrifice of accuracy.
+ lack of theoretical motivation; current theoretical analysis does not provide much insight
+ unclear insights from case study and whether they are attributed to SOP or a combination of a priori knowledge and model outcomes.

**Justification For Why Not Higher Score:**

lack of sufficient evidence for gains in explainability in exchange for compromised accuracy

**Justification For Why Not Lower Score:**

NA

---

### Decision · Program_Chairs · 2024-01-16

Reject